# Preparation and Characterization of Monoclonal Antibodies with High Affinity and Broad Class Specificity against Zearalenone and Its Major Metabolites

**DOI:** 10.3390/toxins13060383

**Published:** 2021-05-27

**Authors:** Yanan Wang, Xiaofei Wang, Haitang Zhang, Hanna Fotina, Jinqing Jiang

**Affiliations:** 1College of Animal Science and Veterinary Medicine, Henan Institute of Science and Technology, Xinxiang 453003, China; wyn564@126.com (Y.W.); histzht@126.com (H.Z.); 2Faculty of Veterinary Medicine, Sumy National Agrarian University, 40021 Sumy, Ukraine; 3Xinke College, Henan Institute of Science and Technology, Xinxiang 453003, China; wangxiaofei1119@126.com

**Keywords:** zearalenone, immunogen, monoclonal antibodies, high affinity and broad class specificity, icELISA, immunoassay

## Abstract

This study aimed to detect and monitor total Zearalenone (ZEN) and its five homologs (ZENs) in cereals and feed. The monoclonal antibodies (mAbs) with a high affinity and broad class specificity against ZENs were prepared, and the conditions of a heterologous indirect competitive ELISA (icELISA) were preliminarily optimized based on the ZEN mAbs. The immunogen ZEN-BSA was synthesized using the oxime active ester method (OAE) and identified using infrared (IR) and ultraviolet (UV). The coating antigen ZEN-OVA was obtained via the 1,4-butanediol diglycidyl ether method (BDE). Balb/c mice were immunized using a high ZEN-BSA dose with long intervals and at multiple sites. A heterologous indirect non-competitive ELISA (inELISA) and an icELISA were used to screen the suitable cell fusion mice and positive hybridoma cell lines. The ZEN mAbs were prepared by inducing ascites in vivo. The standard curve was established, and the sensitivity and specificity of the ZEN mAbs were determined under the optimized icELISA conditions. ZEN-BSA was successfully synthesized at a conjugation ratio of 17.2:1 (ZEN: BSA). Three hybridoma cell lines, 2D7, 3C2, and 4A10, were filtered, and their mAbs corresponded to an IgG1 isotype with a κ light chain. The mAbs titers were between (2.56 to 5.12) × 10^2^ in supernatants and (1.28 to 5.12) × 10^5^ in the ascites. Besides, the 50% inhibitive concentration (IC50) values were from 18.65 to 31.92 μg/L in the supernatants and 18.12 to 31.46 μg/L in the ascites. The affinity constant (*Ka*) of all of the mAbs was between 4.15 × 10^9^ and 6.54 × 10^9^ L/mol. The IC50 values of mAb 2D7 for ZEN, α-ZEL, β-ZEL, α-ZAL, β-ZAL and ZAN were 17.23, 16.71, 18.27, 16.39, 20.36 and 15.01 μg/L, and their cross-reactivities (CRs, %) were 100%, 103.11%, 94.31%, 105.13%, 84.63%, and 114.79%, respectively, under the optimized icELISA conditions. The limit of detection (LOD) for ZEN was 0.64 μg/L, and its linear working range was between 1.03 and 288.55 μg/L. The mAbs preparation and the optimization of icELISA conditions promote the potential development of a rapid test ELISA kit, providing an alternative method for detecting ZEN and its homologs in cereals and feed.

## 1. Introduction

Food and feed contamination with mycotoxins pose a serious threat to human health and animal husbandry development and has caused widespread concern [1]. Zearalenone (ZEN), a mycotoxin, is a toxic secondary metabolite produced by certain species of the genus Fusarium, such as *F**usarium graminearum*, *F**usarium culmorum*, *F**usarium tricinctum*, *F**usarium roseum*, *Fusarium oxysporum*, *F**usarium moniliforme* and *F**usarium semitectum*. ZEN, 6-(10-hydroxy-6-oxycarbenyl)-β-ryanoic acid-μ-lactone, also known as the F-2 toxin, mainly contaminates cereals, including corn, wheat, barley, rice, and oats or foods containing these cereals. Fusarium produces ZEN under natural conditions. ZEN is metabolized or is reduced to its metabolites, including zearalanone (ZAN), α-zearalanol (α-ZAL), β-zearalanol (β-ZAL), α-zearalenol (α-ZEL), and β-zearalenol (β-ZEL), in the body (the molecular structures are shown in Figure 1) [2,3]. ZEN is a non-steroidal estrogenic compound with a toxic estrogen effect, destroying the reproductive system of animals, such as the estrogen syndrome in pigs, despite its low toxicity after oral administration [4,5]. Meanwhile, ZEN also has genetic toxicity, immunotoxicity, endocrine toxicity, and carcinogenic toxicity [6,7,8]. Therefore, ZEN is a key target during cereal food and feed quality and safety monitoring. Most countries have implemented the maximum residue limits (MRLs) of ZEN in cereal foods and feed. For instance, the MRL of ZEN in cereals and cereal products is 2 mg/kg, in corn by-products it is 3 mg/kg, and it is 0.1 mg/kg in compound feeds for piglets and young sows, based on EU guidelines. [9]. In Italy, the MRL of ZEN in cereals and cereal products is 0.1 mg/kg [10] and it is 0.05 mg/kg in Australia [11]. The current ZEN MRL theme in China is “GB 2761-2017 Food Mycotoxin Limit”, strictly showing that the MRL of ZEN in wheat, wheat flour, corn, and cornflour is 0.06 mg/kg [12]. However, researchers have extensively explored the occurrence and toxicity of ZEN and its metabolites due to increased food safety awareness. *Fusarium* spp.-infected cereals contain ZEN, α-ZEL, and β-ZEL. Notably, ZEN is metabolized into α-ZEL, β-ZEL, and ZAN. α-ZAL is metabolized into β-ZAL and ZAN, indicating that both ZEN and its metabolites are toxic to the human body. However, α-ZEL has the highest toxicity, 10–20 times higher than ZEN toxicity [13,14]. Therefore, immense research on detecting total ZEN and its metabolites (TZEN) is necessary since single ZEN detection cannot meet food and feed industry requirements.

Currently, physicochemical analysis and immunoassay are the two main methods used for TZEN detection. The major physicochemical analysis methods used in all countries include thin-layer chromatography (TLC) [15], high-performance liquid chromatography (HPLC) [16], gas chromatography-mass spectrometry (GC-MS) [17], and liquid chromatography/tandem mass spectrometry (LC-MS/MS) [18]. However, these techniques are expensive, time-consuming, and require complex sample pretreatment procedures, expensive instruments, and skilled technicians, making them unsuitable for high-throughput detection [19]. The immunoassay method that is based on the specificity and sensitivity antigen-antibody reaction is a hotspot in the TZEN detection research due to its strong selectivity and sensitivity, high speed, easy sample screening, and large scale on-site operation [20,21]. In recent years, different immunoassay methods based on ZEN monoclonal antibodies (mAbs) with a high affinity and broad class specificity have been established to rapidly detect TZEN. These include an enzyme-linked immunosorbent assay (ELISA) [22,23,24], a gold immunochromatographic assay (GICA) [25], and a fluorescence polarization immunoassay (FPIA) [26]. However, these immunoassay methods have some drawbacks, such as poor specificity and sensitivity to TZEN, possibly due to a low quality mAb, thus not meeting the actual detection needs. Notably, high-quality mAbs are integral to immunoassay methods since immunoassay efficacy depends on the affinity and specificity of the mAbs used. Recent reports have shown that while most mAbs can specifically recognize ZEN, they only recognize some ZEN metabolites [27,28]. However, mAb sensitivity should be further improved, despite the broad specificity [22,23].

The preparation of broad spectrum class specific antibodies is essential for the development of a multiple analogues immunoassay [29,30]. In recent years scholars have researched antigen design and synthesis to broaden the recognition spectrum of antibodies. Thus far, the following three approaches have been used to obtain a broad spectrum of specific antibodies via antigen design and synthesis: (1) The development of a general structural immunogen, with the general structure of the analytes as the detection target, such as aflatoxins (AFs) [31], sulfonamides (SAs) [32], organophosphorus (OPs) [33], and fluoroquinolones (FQs) [34]. (2) The development of a multi-hapten immunogen by simultaneously coupling several different haptens to one carrier protein, such as avermectins (AVMs) [35], microcystins (MCs) [36], and pesticides, including chlorpyrifos, triazophos, carbofuran, and parathion methyl [29]. (3) The development of several mixed immunogens by individually coupling each hapten to a carrier protein. The mixed immunogens, such as tetracycline (TCS) [37] and *Bacillus thuringiensis* (BTS), are then used for simultaneous immunization [38]. However, similar to most mycotoxins, antibiotics, and synthetic drugs, ZEN and its structural analogues are too small to elicit an immune response in the host animal. Therefore, developing a ZEN hapten and conjugating it to a carrier protein is crucial in developing a desirable immunogen. Several strategies of developing ZEN immunogens, including the oxime active ester method (OAE), the formaldehyde method (FA), the 1,4-butanediol diglycidyl ether method (BDE), and the amino glutaraldehyde method (AGA), exist in the literature. However, none of these methods can produce antibodies with a high affinity and broad class specificity for TZEN. For instance, the FA method produces antibodies with a lower affinity [39], while the BDE method-antibodies have a high specificity to ZEN and ZON, except for α-ZAL, β-ZAL, α-ZEL, and β-ZEL [40]. The AGA method-antibodies are highly specific to ZEN and cannot recognize other ZENs [41]. Fortunately, the OAE method-antibodies have a broad specificity against ZENs. Therefore, this study used the OAE method to further improve the quality of the antibodies.

The immune response not only depends on the physicochemical properties of the immunogen, but is also closely related to the immunization method [42]. Animals immunized with the same immunogen can produce antibodies with significantly different immunological properties due to complex immunochemistry. The influencing factors of the immunization method include the injection route, dosage, interval, adjuvant usage, frequency, and individual animal differences, with the dosage and the interval being the key factors. Some scholars have shown that a low immunogen dose induces a narrow-spectrum specificity of antibodies, while a high dose has a broad-spectrum of specificity of antibodies [43]. An immunogen elicits the generation of memory B cells, which require certain immunization intervals to clone and proliferate, producing several specific antibodies. Moreover, a certain immunization interval improves the affinity of antibodies. Generally, the immunization interval in animals is not less than two weeks [42,44]. A substantial dose (200 μg/mL) and a longer (4 weeks) immunization interval were adopted in this study.

The screening of hybridioma cells is a key step in the preparation of high quality mAbs. As a typical response to an artificial hapten-carrier conjugate, B lymphocytes generate different antibody molecules, recognizing different parts of the conjugate. Besides, each B lymphocyte produces a specific type of antibody molecule [45]. Therefore, many immunoassays can be optimized to achieve a higher sensitivity using heterologous coating antigens [32,36]. A heterologous indirect non-competitive ELISA (inELISA) and a heterologous indirect competitive ELISA (icELISA) were employed to select the cell fusion mice and the positive hybridioma cells.

In this study, a new immunogen was designed and synthesized using the OAE method. The ZEN mAbs with a high affinity and a broad class specificity were produced through animal immunization and hybridoma technology. The icELISA protocols based on the selected ZEN mAbs were then optimized, providing technical support for the development of ELISA kits and other immunoassays of TZEN in food and feed.

## 2. Results and Discussion

### 2.1. Immunogen Identification

Infrared spectroscopy (IR) can identify compounds by characterizing their molecular structures and functional groups. Since the immunogen ZEN-BSA is a new compound that is formed by coupling ZEN and bovine serum albumin (BSA), IR is used to characterize whether the new compound contains the characteristic structures of ZEN and BSA, so as to determine whether the synthesis of the new compound is successful. The IR of the immunogen ZEN-BSA synthesized using the OAE method was compared with the IR of BSA. The IR absorptions were similar, between 1500 and 1700 cm^−1^, as evidenced by the peaks produced by the amine group in BSA (Figure 2), indicating the presence of BSA in the artificial immunogen ZEN-BSA. In addition, the IR of the immunogen ZEN-BSA and ZEN had two similar absorptions in the regions 2100–2300 cm^−1^ and 1250–1350 cm^−1^, which are the characteristic peaks produced by hydroxyl, and esters in ZEN. This is an indication that the immunogen ZEN-BSA contains ZEN. BSA does not exhibit any absorption in the same areas. Our findings imply a successful synthesis of ZEN-BSA.

BSA has a characteristic ultraviolet spectrum (UV) absorption peak at 278 nm, and ZEN has characteristic peaks at 236, 274, and 316 nm. The immunogen ZEN-BSA prepared using the OAE method showed the characteristic absorption peaks of both BSA and ZEN, or either of them (green curve in Figure 3), which indicates that the immunogen ZEN-BSA was successfully synthesized. The calculated molar ratio of ZEN to BSA was 17.2:1, based on the Lambert–Beer law.

### 2.2. Selection for Potential Cell Fusion in Immunized Mice

An inELISA and an icELISA were used to measure the titers, the 50% inhibitive concentration (IC50) values, and the cross-reactivities (CRs, %) of ZEN polyclonal antibodies (pAbs) from the five immunized mice after the five inoculations. Although the five immunized mice had a better positive immune response with high titers (>10^3^) (Figure 4), the fifth (No. 5) mouse had the most efficient immune response, with titers of 1:(6.4 × 10^3^). The inhibition curves (Figure 5) showed that the fifth mouse had the lowest IC50 value (45.0 μg/L). The five mice showed different levels of broad-class specificity towards the ZENs, with the fifth mouse having the broadest CRs (>88%) (see Table 1). Thus, the fifth mouse was chosen for the cell fusion experiment.

### 2.3. Screening of Positive Clones and Establishment of Hybridoma Cell Lines

The growing hybridoma cell clones were observed 12 days after the cell fusion. A total of 334 of the 384 wells of four 96-well cell culture plates had hybridoma cells, at a fusion rate of about 87%. An inELISA and an icELISA were used to screen the culture supernatants of all of the wells simultaneously. A total of 57 wells were positive at a rate of about 17%. After subcloning thrice using a limiting dilution, the three stable hybridoma cells, 2D7, 3C2, and 4A10, that had a high-titer and a lower IC50 value in relation to ZEN were used to produce mAbs. The titers and the IC50 values of the mAbs from culture supernatants and ascites are shown in Table 2.

In addition, it is worth noting that the existing mAbs preparation methods include the in vivo-induced ascites method, cell culture medium in vitro culture method, bioreactor in vitro production method, single cell technology and phage display technology, etc. Due to its advantages of simplicity, low cost, short cycle, high antibody concentration and high yield, the in vivo-induced ascites method is still widely used in most laboratories. Yet, the methodology remains debatably acceptable. For example, the European Union Reference Laboratory for alternatives to animal testing (EURL ECVAM) has clearly stated that “Antibody production using the ascites method should no longer be acceptable under any circumstances” (Barroso et al. EURL ECVAM Recommendation on Non-Animal-Derived Antibodies). Although the in vivo ascites induction method was still used in this study, the authors suggest that the in vitro culture method should be advocated in future studies instead of the in vivo ascites induction method for the production of mAbs for the consideration of animal protection and animal ethics.

### 2.4. Identification of the Karyotype, Isotype, and Stability of Hybridoma Cell Lines

The chromosomes of myeloma NS0 cells were between 62 and 68, and those of mouse spleen cells were 40. However, hybridoma cell chromosomes were between 96 and 102 (mean, 98.4), indicating that the cell lines fused by the two parents were the hybridoma cells since their chromosomes were more than those of the individual parents. Additionally, a mouse mAb isotyping kit was used to determine that all the three mAbs belonged to the IgG1 isotype with a κ light chain. Stability verification showed that the three hybridoma cell lines secreted mAbs. The titer and IC50 values of ZEN mAbs against ZEN in the supernatants of each generation were determined after five repeated freeze-thaw cultures, but there was no significant difference. Besides, the mAb-secreting hybridoma cells were genetically stable (Figure 6 and Figure 7).

### 2.5. Determination of ZEN mAbs’ Affinity

The antibody affinity reflects its binding strength with an antigen. The affinity constant (*Ka*) represents the antibody affinity. The equilibrium dialysis method, the Scatchard diagram method, the Batty saturation method, the competitive binding method, and the precipitation method are the commonly used to determine *Ka*. In this study, the Batty saturation method, a simple, rapid, and reliable method, was used. The *Ka* of 2D7, 3C2, and 4A10 were 6.54 × 10^9^, 5.32 × 10^9^, and 4.15 × 10^9^ L/mol, respectively, (Figure 8), indicating that 2D7 had the highest affinity.

### 2.6. Assessment of the Sensitivity and Specificity of ZEN mAbs

Generally, sensitivity and specificity are key in assessing the quality of antibodies since they determine immunoassay performance. Sensitivity is one of the important indicators for evaluating the antigen–antibody reaction mode, that is, the immunoassay method, represented by an IC50 value. In this study, icELISA conditions, such as the best working concentration of coating antigen and antibody, concentration of organic solvent, ionic strength, and pH value, were optimized (detailed results are provided in the section titled “Optimisation of the icELISA”). Therefore, the standard curve of an icELISA was established, and the IC50 and CRs values of the antibodies were determined (Table 3). The IC50 values of mAb 2D7, mAb 3C2, and mAb 4A10 were from 15.78 to 21.41 μg/L, 24.79 to 29.22 μg/L, and 31.46 to 41.61 μg/L, respectively, indicating a lack of significant difference. However, mAb 2D7 had the highest sensitivity to six ZENs (ZEN, α-ZAL, β-ZAL, α-ZOL, β-ZOL and ZON).

Specificity is the ability of an antibody to recognize a paired antigen or hapten and is determined by the complementarity between the spatial structure of the hypervariable region of an antibody molecule and the antigenic determinant. A CR value usually expresses specificity. The CR values of three ZEN mAbs, mAb 2D7, mAb 3C2, and mAb 4A10, were from 84.63 to 114.83, 86.82% to 102.34%, and 75.61 to 100%, respectively, indicating a lack of significant difference (Table 4). However, mAb 2D7 had the highest CR value. Since mAb 2D7 had the lowest IC50 value and the broadest CR value in relation to the six ZENs, it was selected for subsequent studies. Meanwhile, the IC50 values of the three mAbs against aflatoxin B1, deoxynivalenol, T-2, toxin and ochratoxin A were all greater than 10000 μg/L and their CR values were all less than 1%, this indicated that the ZEN mAbs had no cross-reaction with other mycotoxins.

Several recent reports have used immunogens developed using the OAE method to obtain antibodies with a broad-specificity and sensitivity to ZENs. A comparison between mAb 2D7 in this study and other ZEN mAbs reported in the recent literature is shown in Table 4. mAb 2D7 had a high affinity with IC50 values between 15.78 and 21.41 μg/L and a high broad class-specificity with CR values between 84.63 and 114.83%, meeting the expectations. Therefore, mAb 2D7 can be used to establish immunoassays for ZEN analysis in cereals, feed, and other samples. However, mAb 2D7 had a lower broad-specificity than kk-ZEN as reported by Cha et al. [22]. In addition, mAb 2D7’s sensitivity was substantially lower than the sensitivity of mAb 3D4 previously reported by Zhang et al. [26], mAb 7B2 reported by Liu et al. [24], mAb 15C2 reported by Pei et al. [46], mAb 2D8 reported by Burmistrova et al. [47], and mAb 2C5 reported by Thongrussamee et al. [48]. In contrast, ZEN antibodies had significantly different immunological properties, such as affinity, sensitivity, and specificity, despite using the same OAE to prepare the ZEN immunogen (see Section 4.5 “Synthesis and Identification of Immunogen and Coating Antigen”). Therefore, it is necessary to consider the physical and chemical characteristics of hapten, the immunogen synthesis method, and the coupling rate, immunization procedure, antibody screening methods, and other factors to prepare broad class b-specific and sensitive antibodies.

### 2.7. icELISA Optimization

Some challenges, such as the optimal working concentration of the antigen and the antibody, the concentration of the organic solvent, ionic strength, and pH value, are often encountered during the establishment of the icELISA method. In this study, the icELISA conditions were optimized to improve its sensitivity. The chessboard titration results indicated that the optimal concentration of the coating antigen was 2 μg/mL, mAb 2D7 was 0.3 μg/mL (1:5000), and GaMIgG-HRP was 0.6 μg/mL (1:1000). Methanol is widely used in an ELISA as an organic cosolvent to extract the mycotoxins from the food and feed matrix. An appropriate methanol level not only affects ELISA sensitivity but also helps to dissolve analytes. The Amax and the Amax/IC50 value decreased when the methanol concentration exceeded 30%, but there was no significant effect when the methanol concentration was below 30% (Figure 9). The effects of the pH value on the icELISA are shown in Figure 10. The pH values between 5.0 and 9.0 had no significant effect on the Amax/IC50 and Amax. However, the Amax/IC50 and Amax were highest at a pH of 7.4, indicating full binding between antibodies and the antigen. Therefore, choosing a PBS with a pH value between 7.2 and 7.4 is the most suitable in an icELISA system.

The standard curve of mAb 2D7 was established under the optimized icELISA conditions for ZEN, as shown in Figure 11. The IC50 values calculated for ZEN, α-ZEL, β-ZEL, α- ZAL, β-ZAL, and ZAN were 17.23, 16.71, 18.27, 16.39, 20.36, and 15.01 μg/L, respectively, and the CRs were 100, 103.11, 94.31, 105.13, 84.63, and 114.79%, respectively. The limit of detection (LOD) (IC15) for ZEN was 0.64 μg/L, and the linear working range (IC20-IC80) for ZEN ranged from 1.03 to 288.55 μg/L.

## 3. Conclusions

The preparation of broad class specific and sensitive antibodies is essential for total ZEN detection. In this study, the OAE method was used to synthesize the immunogen ZEN-BSA at a conjugation ratio of 17.2:1. Balb/c mice were immunized with a high dose at long intervals and multiple sites. Besides, a heterologous ELISA and an icELISA were used to screen suitable cell fusion mice and positive hybridoma cells. This study has generated ZEN mAbs with a high affinity and broad class specificity capable of recognizing ZEN and its five homologs. Further, the conditions of a heterologous icELISA have been optimized, providing a basis for the establishment of ELISA kits and other immunoassay methods.

## 4. Materials and Methods

### 4.1. Chemicals and Reagents

ZEN standard (solvent-free), α-ZAL, β-ZAL, α-ZEL, β-ZEL, and ZAN standard solutions in methanol, O-carboxymethoxylamine hemihydrochloride (CMO), 1-ethyl-3-(3-dimethylaminopropyl) carbodiimide (EDC), N,N′-Dicyclohexylcarbodiimide (DCC), N-hydroxysuccinimide (NHS), polyethylene glycol 1500 (PEG 1500, 50%), hypoxanthine aminopterin thymidine (HAT), hypoxanthine thymidine (HT), phenacetin, 3,3,5,5-tetra-methylbenzidine (TMB), urea peroxide, and Tween-20 were sourced from Sigma-Aldrich (St. Louis, MO, USA). Bovine serum albumin (BSA), ovalbumin (OVA), Freund’s complete adjuvant (FCA), Freund’s incomplete adjuvant (FIA), and culture media RPMI-1640 with L-glutamine were obtained from Pierce (Rockford, IL, USA). Goat anti-mouse IgG conjugated with horseradish peroxidase (GaMIgG-HRP) and a mouse mAb isotyping kit were acquired from Sino-American Biotechnology Company (Shanghai, China). Fetal bovine serum (FBS) was sourced from Hangzhou Sijiqing Biological Engineering Materials Co., Ltd. (Hangzhou, China). Besides, 1,4-Butanediol diglycidyl ether (BDE), pyridine, dioxane, and dimethylformamide (DMF) were obtained from J&K Chemicals (Shanghai, China). Ethylenediamine (EDA) was also sourced from J&K Chemicals (Shanghai, China). All other solvents, reagents, and chemicals were standard commercial products of analytical grade or better. Cell culture plates (6, 24, and 96 wells) and culture flasks were obtained from Costar Inc. (Bethesda, MD, USA). Transparent 96-well polystyrene microtiter plates were sourced from Boyang Experimental Equipment Factory (Jiangsu, China).

### 4.2. Standard Solutions, Buffers, and Hybridoma Growth Media

The standard solutions included the following: (1) stock solutions of ZEN, α-ZAL, β-ZAL, α-ZOL, β-ZOL, and ZON had a concentration of 1 mg/mL, were prepared using methanol, and diluted to a standard solution with 0.01 mol/L PBS. (2) To create the cationized carrier protein solution, 200 mg (0.003 mmol) of BSA (or 135 mg, 0.003 mmol, OVA) was dissolved in 10 mL of PBS. To this, 7.2 mg (0.12 mmol) of EDA and 11.5 mg (0.06 mmol) of EDC were then added sequentially and magnetically stirred at room temperature for 2 h. The reaction solution was dialyzed using PBS at 4 °C for three days. The dialysate was changed once a day to obtain cationized BSA (cBSA) or cOVA, then stored at 4 °C for subsequent analysis.

The buffers included the following: (1) A phosphate buffer saline (0.01 M PBS, pH 7.4) comprising NaCl (137 mmol), Na_2_HPO_4_·12H_2_O (10 mmol), KCl (2.68 mmol), and KH_2_PO_4_ (1.47 mmol). (2) A carbonate buffer (0.05 M CBS, pH 9.6) comprising Na_2_CO_3_ (15 mmol) and NaHCO_3_ (35 mmol) used as a coating buffer, and (3) The washing buffer was PBS containing 0.05% Tween-20 (PBST). (4) A blocking buffer containing swine serum (5%, *v/v*) in PBST and (5) The substrate buffer was a mixture of part A (500 mL) and part B (500 mL) solutions. Part A contained (per 1 L of water) 3.15 g citric acid, 6.966 g anhydrous sodium acetate, 0.08 g phenacetin, and 0.05 g urea peroxide adjusted to a pH of 5.0 using HCl; and Part B had 1.27 g of TMB dissolved in 500 mL of methanol and 500 mL of glycerol. (6) A 2 M H_2_SO_4_ was used as the stopping solution.

The hybridoma growth media were as follows: (1) A complete medium consisting of 78 mL of RPMI-1640 medium, 20 mL of FBS, 1 mL of antibiotics, and 1 mL of HEPES. (2) A cell freezing solution with dimethyl sulphoxide (DMSO, 10%, *v/v*) in complete medium.

### 4.3. Equipment and Instruments

A spectrophotometric microtiter reader (MULTISKAN MK3, Thermo Co., Shanghai, China) was used to measure the absorbance. A Bruker Tensor 27 spectrometer (Bruker Optics Inc., Bruker, Germany) was used for the infrared (IR) spectra analysis, and a DU-800 UV–visible spectrophotometer (Beckman-Coulter, Fullerton, CA, USA) was used for the ultraviolet (UV)–visible spectra analysis. A Galaxy S-type CO_2_ incubator (RS-Biotech, Ayrshire, UK) was used for cell cultivation, and a TS100-F inverted microscope (Nikon Company, Tokyo, Japan) was used for cell observation. Moreover, a 303A-1 electric heating constant temperature incubator was sourced from Beijing Zhongxing Weiye Instrument Co., Ltd. (Beijing, China), and an Exceed DZG-303A ultrapure water polishing system was obtained from Chengdu Kangning Special Experiment Pure Water Equipment Factory (Chengdu, China). An LDZX-30 KB vertical pressure steam sterilizer was acquired from Shanghai Shenan Medical Instrument Factory (Shanghai, China), and a SW-CJ-2 FD superclean bench was obtained from Suzhou Purification Equipment Co., Ltd. (Suzhou, China).

### 4.4. Experimental Animals and Cells

The six-week-old female Balb/c mice and their feed were obtained from the Experimental Animal Center of Medical College of Zhengzhou University (Zhengzhou, China). The animals were housed at 24 ± 2 °C, and a relative humidity of 50 ± 20%, with a 12 h light/dark cycle. Diet-wise, they were given tap water and fed ad libitum. NS0 myeloma cells were obtained from the Key Laboratory of Animal Immunology of the Ministry of Agriculture (Zhengzhou, China).

### 4.5. Synthesis and Identification of Immunogen and Coating Antigen

The immunogen ZEN-BSA was synthesized using a modified oxime active ester method (OAE), as previously described [22,23]. Briefly, 5 mg (0.0157 mmol) ZEN was dissolved in 2 mL of pyridine. To this, 10 mg (0.09117 mmol) of CMO was then added while stirring using a magnetic bar at room temperature for 24 h to obtain the yellow solution product. The product was dried using nitrogen. Then, 3 mL of deionized water was added, and the pH was adjusted to 8.0 using 0.1 mol/L NaOH. The product was extracted thrice using an equal volume of ethyl acetate. The aqueous phase was discarded, and the organic phase was collected and dried using nitrogen to obtain hapten ZENO (light yellow oil). ZENO was dissolved in 2 mL of dioxane. Then, 2.5 mg of NHS and 5 mg of DCC were added and stirred at 4 °C for 4 h to prepare the hapten activation solution. Next, 1 mL of cBSA activation solution was then added dropwise at 20 mg/mL, stirred at 4 °C for 4 h, then dialyzed using PBS at 4 °C for three days. The dialysate was changed once a day and stored at 4 °C for subsequent analysis. The synthetic route of ZEN-BSA (OAE) is shown in Figure 12.

The coating antigen ZEN-OVA was prepared using a modified 1,4-butanediol diglycidyl ether method (BDE) [40]. Briefly, 5 mg (0.0157 mmol) of ZEN was dissolved in 0.5 mL of DMF. Then, 31 μL (0.157 mmol) of BDE was dissolved in 0.5 mL of double-distilled water and added dropwise to the ZEN solution. The pH was adjusted to 10.8 using 1 mol/L NaOH and magnetically stirred at room temperature for 4 h to obtain the hapten activation solution. The remainder of the procedure was similar to the ZEN-BSA procedure, except that OVA replaced BSA. The synthetic route is shown in Figure 13.

Ultraviolet (UV) scanning spectroscopy was used to confirm the synthesis of the immunogen ZEN-BSA. The molecular binding ratio of Zen to BSA was calculated based on Lambert–Beer law, A = εCL (where A is the absorbance value, ε is the molar extinction coefficient, (constant value), C is the solute concentration, and L is the optical path).

### 4.6. ZEN mAb Preparation

#### 4.6.1. Animal Immunization

Five six-week-old female Balb/c mice were immunized using a high ZEN-BSA conjugate dose at long intervals with multiple-site subcutaneous injections [22,43]. For the first immunization, each mouse was subcutaneously injected on four sites (on the back) with 200 μg of the ZEN-BSA conjugate dissolved in sterilized PBS and emulsified with an equal volume of FCA. Four weeks later, booster injections were administered four times at four-week intervals with the same dose in FIA. Four weeks after the last immunization, the tail-amputated blood was collected and separated to obtain the antisera (ZEN pAb) for each mouse. The heterologous inELISA was used to measure the ZEN pAb titer (immunoreactivity). Furthermore, the 50% inhibitive concentration (IC50, represents the sensitivity of the ZEN pAb), and the cross-reactivity test (CR%, indicates the specificity of the ZEN pAb) was determined using a heterologous indirect competitive ELISA (icELISA). Four days before cell fusion, the mouse with the highest ZEN pAb titer, lowest IC50 value, and significant CR was administered an intraperitoneal booster injection of 200 µg of ZEN-BSA conjugate without any adjuvant. The mouse was sacrificed, and the spleen was harvested to obtain hybridomas.

#### 4.6.2. Cell Fusion, Cloning, and Establishment of ZEN mAb Hybridoma Cell Lines

Cell fusion and screening of positive hybridoma cell lines were performed using common operation methods [49] with some modifications. Briefly, the splenocytes were isolated and fused with NS0 myeloma cells at a 10:1 ratio using PEG 1500 as a fusing agent. The fused cells were put into 96-well culture plates where mouse peritoneal macrophages were prepared as feeder cells from young Balb/c mice the day before and grown with a selective HAT medium. After 10 to 14 days, an inELISA and an icELISA were used to screen the positive hybridoma colonies obtained from supernatants. The positive clones were then transferred to 24-well plates to culture. After seven days, positive hybridomas were subcloned thrice using the limiting dilution method. The colonies of interest were then frozen in a culture medium containing 10% DMSO, cryopreserved in liquid nitrogen, then defrosted thrice to screen and identify the stable hybridoma cell lines.

#### 4.6.3. ZEN mAb Production and Purification

The in vivo-induced ascites method was used to produce several ZEN mAbs [50]. Briefly, a mature female Balb/c mouse was intraperitoneally injected with 0.8 mL of paraffin ten days before receiving an intraperitoneal injection of the positive hybridoma cells (1 to 5 × 10^6^ cells). Ascites fluid was collected two weeks later. The saturated ammonium sulfate precipitation method was used to purify the ZEN mAb and then stored at −20 ℃ for subsequent analysis.

### 4.7. ZEN mAb Characterization

#### 4.7.1. Analysis of Karyotype, Isotype, and Stability of Hybridoma Cell Lines

The colchicine blocking method was used to identify Karyotype [51]. A commercially available mouse mAb isotype kit was used to determine the class and subclass of each mAb isotype. The cryopreserved hybridoma cells were resuscitated and passaged once every ten days, five times. An inELISA and an icELISA were used to detect the antibody titers and IC50 values of ZEN in the supernatants at different passages to determine the stability of the hybridoma cells secreting an antibody.

#### 4.7.2. Identification of Affinity, Sensitivity, and Specificity of ZEN mAbs

The Batty saturation method was used to determine the affinity, as follows [52]:*Ka* = (*n* − 1)/[2(*n*[Ab’]t − [Ab]t)](1)
where *n* = [Ag]/[Ag’], [Ag]t and [Ag’]t indicate the different concentrations of a coating antigen, and [Ab]t and [Ab’]t represent the corresponding 50% Amax value of the ZEN mAb concentration when the coating antigen is of different concentrations. An icELISA was used to determine the IC50 value of the ZEN mAb, (sensitivity). The CR (%) described by Ertekin et al. [53] was performed to analyze the specificity of ZEN mAbs, ZEN and its five homologs, including α-ZAL, β-ZAL, α-ZEL, β-ZEL, and ZAN. Furthermore, common mycotoxins, including aflatoxin B1 (AFB1), deoxynivalenol (DON), T-2 toxin (T-2), and ochratoxin A (OTA), were selected as inhibitors in CR. The CR was calculated as follows:CR (%) = [IC50 (ZEN)/IC50 (competitor)] × 100%.(2)

### 4.8. ELISA Procedures

The inELISA procedure was similar to that described in a previous study [54]. Briefly, the coating antigen ZEN-OVA was diluted in CBS at 2 μg/mL, added to the 96-well microplate at 100 μL/well, and incubated at 37.8 °C for 2 h. The microplate was washed thrice using PBST, and unbound active sites were blocked with 250 µL/well of blocking buffer at 37.8 °C for 1 h or at 4 °C overnight. The microplate was then washed, and 50 μL/well of ZEN antibody at an appropriate dilution was added and incubated at 37.8 °C for 15 min. The microplate was washed again, and GaMIgG-HRP (50 μL/well) was added, then incubated at 37.8 °C for 30 min. The microplate was washed six times and 60 µL/well was freshly prepared. The TMB solution was added, then incubated at room temperature for 10 min. The reaction was stopped by adding 2 mol/L H_2_SO_4_ (100 μL/well), then the absorbance was measured at 450 nm. Pre-immunization of serum and PBST were used as negative and blank controls, respectively. The ZEN antibody titers were calculated as the reciprocal of the dilution, causing an absorbance value twice that of the blank value.

Similarly, the icELISA protocol was similar to that of the inELISA, except for the introduction of the competition step after blocking by adding 50 μL/well of standards or analytes, then 50 μL/well of appropriate antibodies. The competitive inhibitory curves were obtained by plotting the concentrations (Log C) versus the B/B0% values (proportion of mean absorbance value of the standards (B) divided by that of the zero standards (B0) in triplicate experiments). The regression equation was then deduced, and the IC50 was calculated. The limit of detection (LOD) and the working range indicated the icELISA sensitivity. The LOD and working range of the icELISA were defined as the minimum ZEN concentration with a 15% inhibition rate (IC15) and a 20–80% inhibition rate (IC20-IC80), respectively, based on the standard curve and regression equation [55].

### 4.9. Optimization of the icELISA Method

A checkerboard titration procedure was used to determine the optimal working concentrations of the coating antigen ZEN-OVA, ZEN mAb, and GaMIgG-HRP. A suitable absorbance (1.0) at 450 nm was selected as the optimal working concentration [56]. The ZEN standard stock solution was diluted in 30% methanol-PBS (30:70, *v/v*) to obtain a seven-point standard curve using the above optimized conditions at various concentrations (1.0, 3.0, 9.0, 27.0, 81.0, 243.0, and 729.0 μg/L) of ZEN standard solutions in an icELISA. The IC50 value indicated the sensitivity, and the CR indicated the specificity of the icELISA method. The ratio of Amax /IC50 was used as the evaluation criteria, and some physicochemical parameters, such as the methanol concentration (10, 20, 30, 40 and 50%, V/V) and pH value (5.0, 6.0, 7.0, 7.4, 8.0 and 9.0) of the assay buffer were tested and optimized to improve the performance of the icELISA method [57].

### 4.10. Data Statistics and Image Processing

Origin Pro 2018 (OriginLab corparation, Northampton, USA) and Excel software (Microsoft Corparation, Redmond, WA, USA) were used for plotting the standard curves and the data analysis. ChemSketch 12.0 (Advanced Chemical Development Co., Ltd., Victoria, Canada) was used to sketch the chemical formulas.

## Figures and Tables

**Figure 1 toxins-13-00383-f001:**
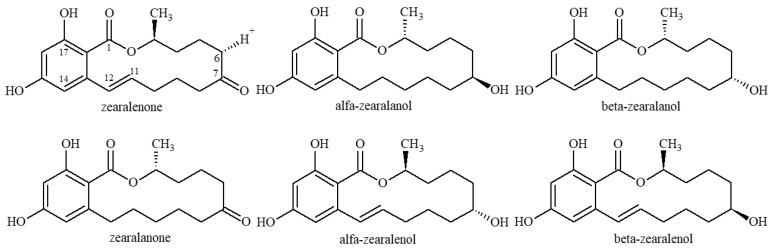
The chemical structure of zearalenone and its metabolites.

**Figure 2 toxins-13-00383-f002:**
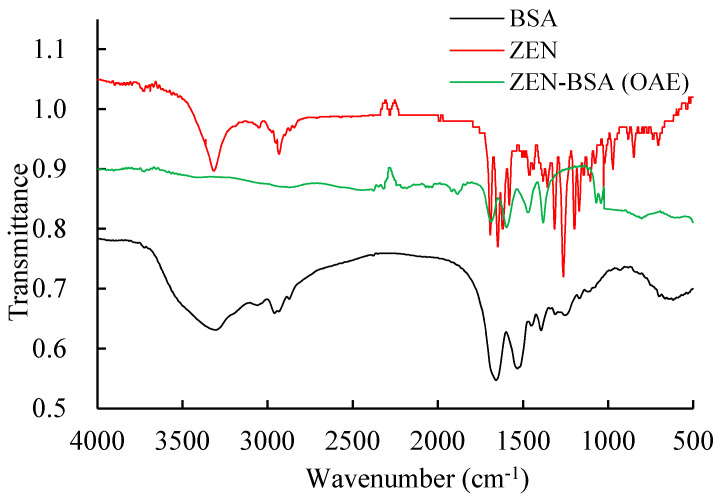
IR spectra of the ZEN-BSA synthesized using the OAE method. IR:Infrared. ZEN-BSA: zearalenone-bovine serum albumin; ZEN: zearalenone; BSA: bovine serum albumin. OAE: oxime active ester method.

**Figure 3 toxins-13-00383-f003:**
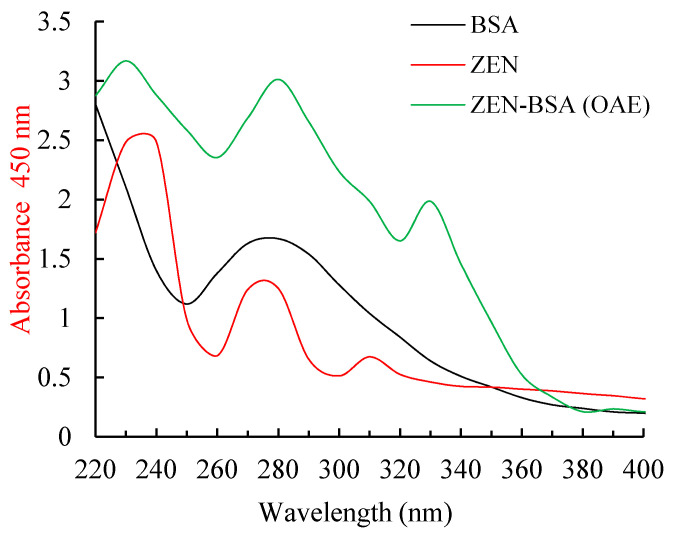
UV spectra of the ZEN-BSA synthesized using the OAE method. UV: ultraviolet.

**Figure 4 toxins-13-00383-f004:**
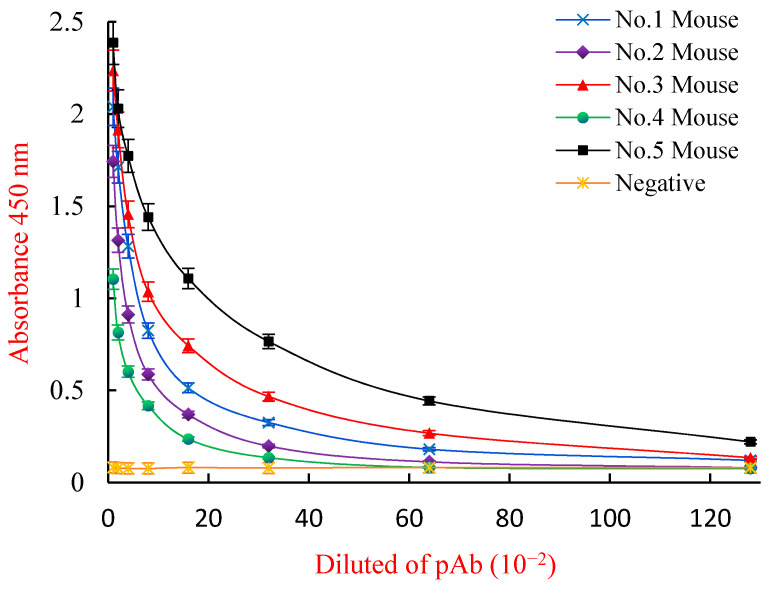
Indirect ELISA titer measurements for the ZEN pAbs. Each point represents the mean of three replicates (*n* = 3). ELISA: enzyme linked immunosorbent assay. ZEN pAbs: zearalenone polyclonal antibodies.

**Figure 5 toxins-13-00383-f005:**
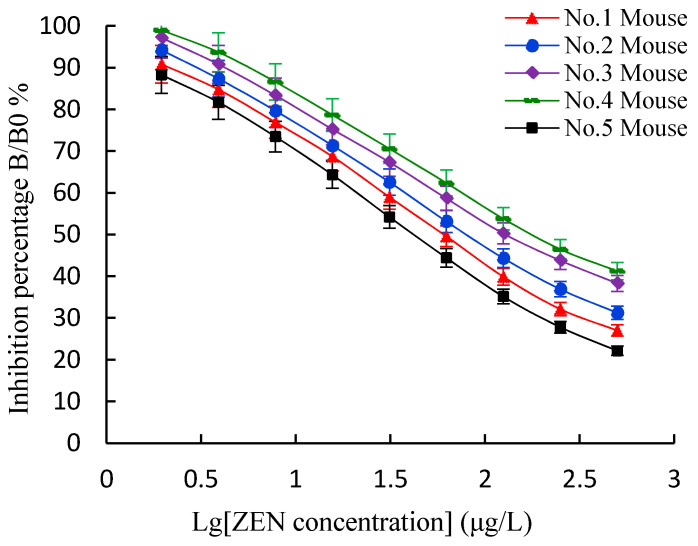
Sensitivity measurement of ZEN pAb to ZEN using an icELISA. The values indicate the mean of three independent assays (*n* = 3).

**Figure 6 toxins-13-00383-f006:**
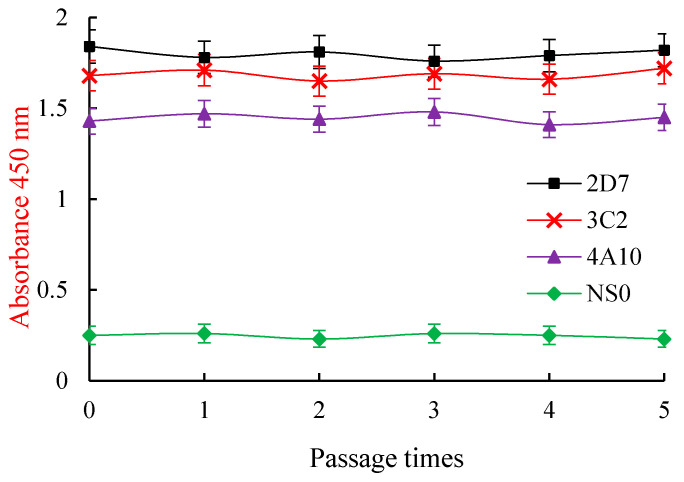
The titers of the ZEN mAbs secreted by three hybridomas after five freeze/thaw cycles. Each point represents the average of three separate assays in triplicate.

**Figure 7 toxins-13-00383-f007:**
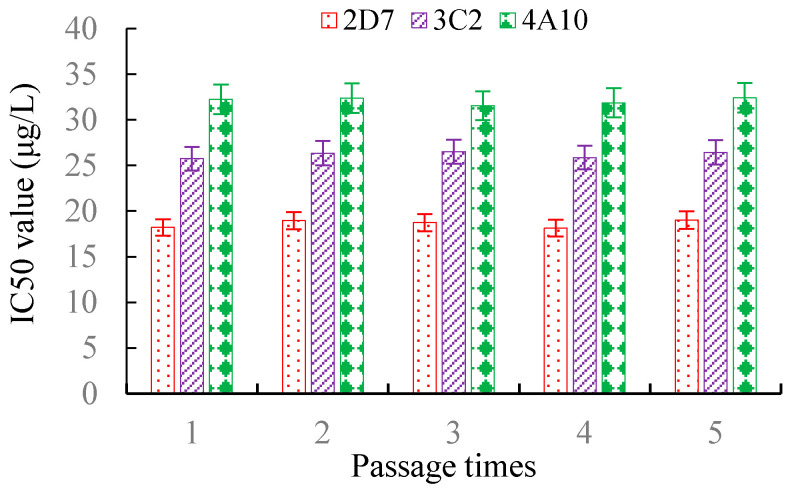
The IC50 values of the ZEN mAbs secreted by three hybridomas after five freeze/thaw cycles. All of the data were calculated from triplicate assays.IC50: 50% inhibitive concentration. ZEN mAbs: zearalenone monoclonal antibodies.

**Figure 8 toxins-13-00383-f008:**
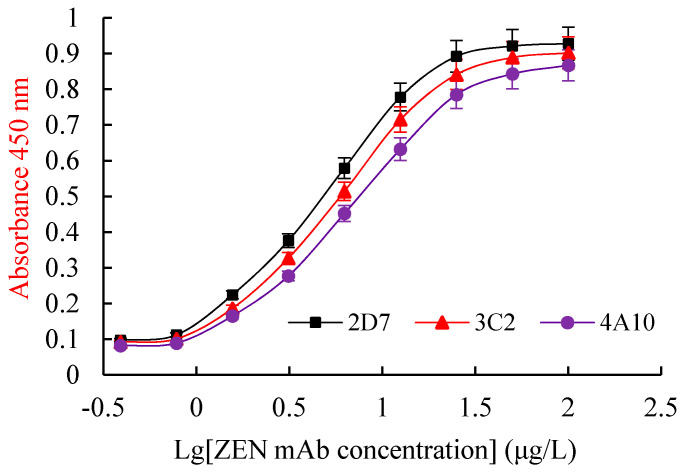
The *Ka* curves of ZEN mAbs. Al of the data were calculated from triplicate assays. *Ka*: affinity constant.

**Figure 9 toxins-13-00383-f009:**
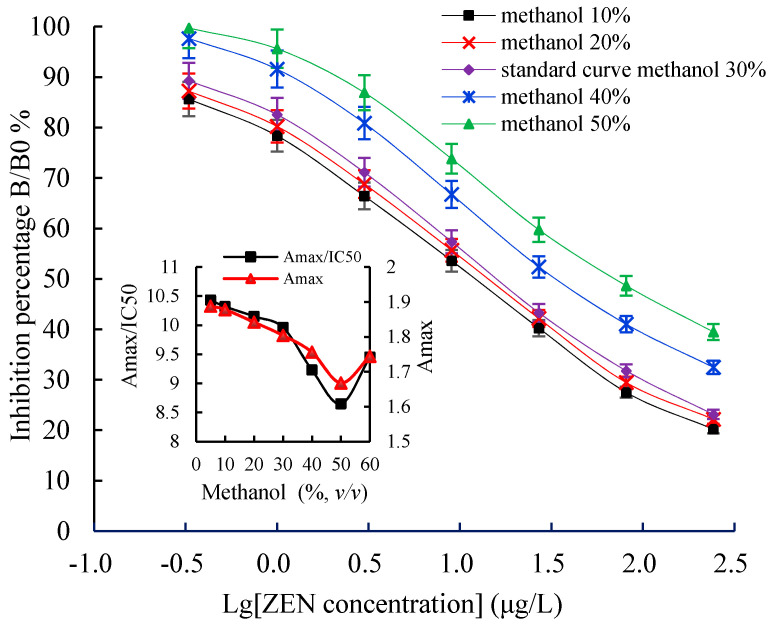
The effects of methanol concentration on icELISA. Each value represents the mean of three replicates. icELISA: indirect competitive enzyme linked immunosorbent assay.

**Figure 10 toxins-13-00383-f010:**
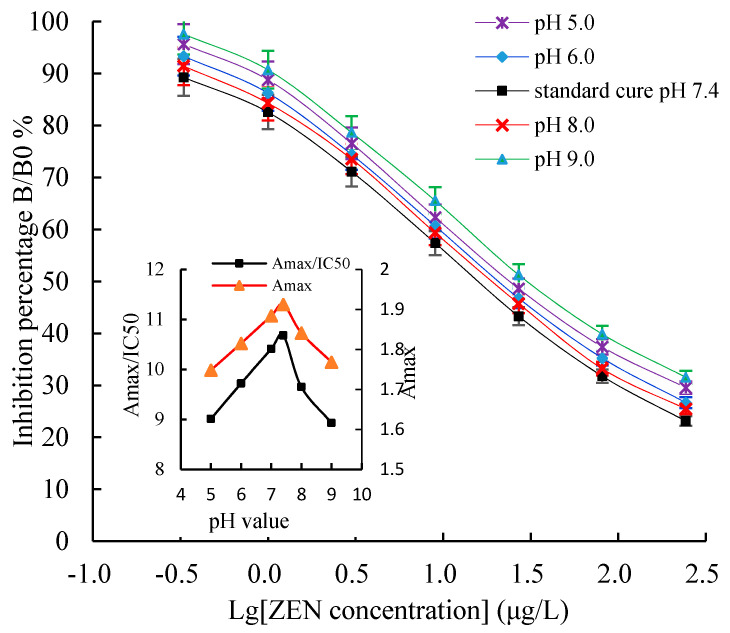
The effects of pH value on an icELISA. Insets indicate the fluctuations of Amax/IC50 (Y-principal axis) and Amax (Y-secondary axis) as a function of pH value. Each value represents the mean of three replicates.

**Figure 11 toxins-13-00383-f011:**
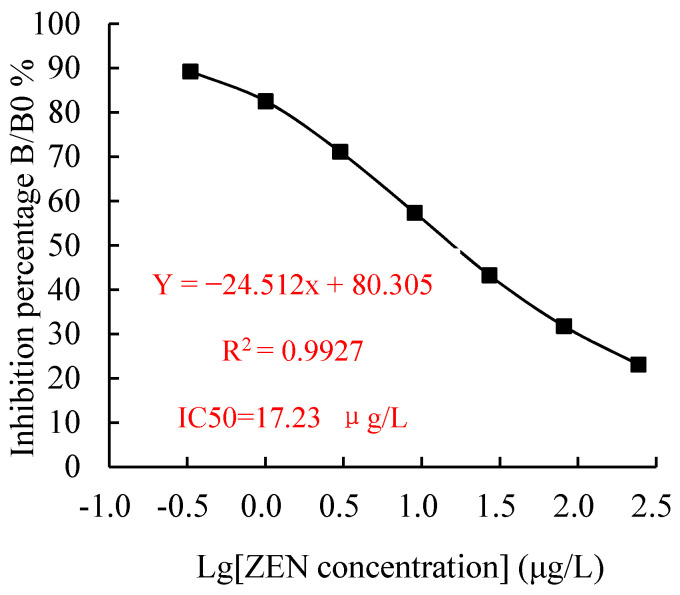
The standard curve of an icELISA for ZEN. The standard curve was obtained using icELISA optimized conditions for ZEN. The data represent the means of three replicates.

**Figure 12 toxins-13-00383-f012:**
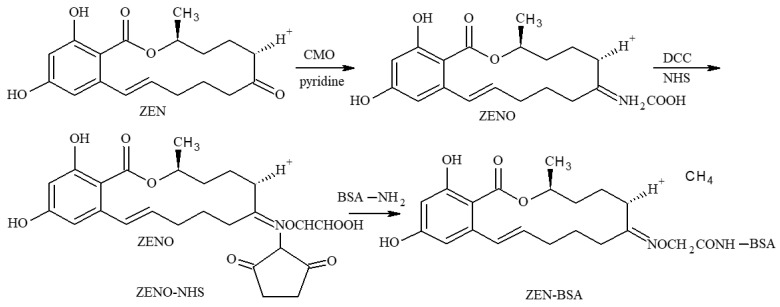
Synthesis route of the ZEN-BSA using the OAE method.

**Figure 13 toxins-13-00383-f013:**
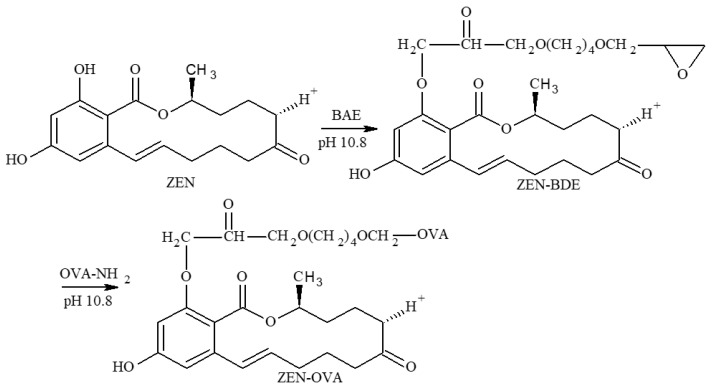
Synthesis route of the ZEN-BSA using the BDE method.

**Table 1 toxins-13-00383-t001:** The cross-reactivity of ZEN pAbs with ZEN and its homologues.

Compound	ZEN pAb (No.1 Mice)		ZEN pAb (No.2 Mice)		ZEN pAb (No.3 Mice)		ZEN pAb (No.4 Mice)		ZEN pAb (No.5 Mice)	
	IC50 (μg/L) ^a,b^	CR (%) ^a^	IC50 (μg/L) ^a,b^	CR (%) ^a^	IC50 (μg/L) ^a,b^	CR (%) ^a^	IC50 (μg/L) ^a,b^	CR (%) ^a^	IC50 (μg/L) ^a,b^	CR (%) ^a^
ZEN	64.1	100	88.3	100	148.3	100	199.4	100	45.0	100
α-ZAL	54.1	118.4	84.8	104.2	170.9	86.7	253.9	78.5	38.6	116.8
β-ZAL	65.2	98.3	95.6	92.4	196.7	75.4	310.2	64.3	47.7	94.4
α-ZOL	61.2	104.7	102.0	86.6	225.2	65.9	339.3	58.8	39.0	111.5
β-ZOL	65.8	97.4	114.0	77.4	238.6	62.1	407.4	48.9	50.9	88.6
ZON	60.5	105.8	95.6	92.4	188.5	78.7	299.9	66.5	37.1	121.5

Note. ^a^ All of the data were calculated from triplicate assays. ^b^ The compound standard solution was prepared in 70% methanol-PBS (7:3, *v/v*). ZEN pAbs: zearalenone polyclonal antibodies.

**Table 2 toxins-13-00383-t002:** The titers and IC50 values of the ZEN mAbs produced by three hybridomas.

ZEN mAbs	Titers of Supernatants ^a^	Titers of Ascites ^a^	IC50 Values of Supernatants (μg/mL) ^a,b^	IC50 Values of Ascites (μg/mL) ^a,b^
2D7	5.12 × 10^2^	5.12 × 10^5^	18.65	18.12
3C2	2.56 × 10^2^	2.56 × 10^5^	26.13	25.37
4A10	2.56 × 10^2^	1.28 × 10^5^	31.92	31.46

Note. ^a^ All of the data were calculated from triplicate assays. ^b^ The ZEN standard solution was prepared in 70% methanol–PBS (7:3, *v/v*). IC50: 50% inhibitive concentration. ZEN mAbs: zearalenone monoclonal antibodies.

**Table 3 toxins-13-00383-t003:** The sensitivity (IC50) and specificity (CR) of three mAbs against ZENs.

Compound	2D7		3C2		4A10	
	IC50 (μg/L) ^a,b^	CR (%) ^c^	IC50 (μg/L) ^a,b^	CR (%) ^c^	IC50 (μg/L) ^a,b^	CR (%) ^c^
ZEN	18.12	100	25.37	100	31.46	100
α-ZAL	17.56	103.19	26.26	96.61	35.86	87.73
β-ZAL	19.21	94.33	28.07	90.38	36.87	85.33
α-ZOL	17.25	105.04	26.66	95.16	35.54	88.52
β-ZOL	21.41	84.63	29.22	86.82	41.61	75.61
ZON	15.78	114.83	24.79	102.34	34.81	90.38
Aflatoxin B1	>10,000	<1	>10,000	<1	>10,000	<1
Deoxynivalenol	>10,000	<1	>10,000	<1	>10,000	<1
T-2 toxin	>10,000	<1	>10,000	<1	>10,000	<1
Ochratoxin	>10,000	<1	>10,000	<1	>10,000	<1

Note. ^a^ All of the data were calculated from triplicate assays, and the average coefficient of variation (CV) was below 10%. ^b^ The compound standard solution was prepared in 70% methanol-PBS (7:3, *v/v*). ^c^ All of the data were calculated using the CR of ZEN mAbs against ZEN as 100%.

**Table 4 toxins-13-00383-t004:** Comparison of IC50 values and CR values of mAb 2D7 and other previously reported ZEN mAbs.

References	mAb	Coupling Reagent	Coupling Rate (ZEN:Carrier)	Immunity Intervals	IC50 of ZEN (μg/L)	CR (%) ^a^				
						α-ZAL	β-ZAL	α-ZOL	β-ZOL	ZON
This study (2021)	2D7	DCC	17.2:1	4 weeks	18.12	103.19	94.33	105.04	84.63	114.83
Dong et al. (2018) [23]	6C2	DCC	14.5:1	2 weeks	114.0	89.5	39.3	99.2	55.5	-
Zhang et al. (2017) [26]	3D4	EDC	-	3 weeks	0.041	107.9	57.2	103.5	73.9	66
Liu et al. (2015) [24]	7B2	DCC	-	4 weeks	0.18	46.7	39.2	60.5	24.7	59.5
Pei et al. (2013) [46]	15C2	DCC	24:1	2 weeks	1.79	-	24.03	188.62	43.77	-
Cha et al. (2012) [22]	Kk-ZEN	EDC	-	2 weeks	131.3	108.1	119.3	114.1	130.3	-
Burmistrova et al. (2009) [47]	2D8	DCC	-	2 weeks	0.8	69	<1	42	<1	22
Thongrussamee et al. (2008) [48]	2C5	DCC	-	2 weeks	1.79	121.5	65.3	21.5	18.9	-

Note. ^a^ All of the data were calculated using the CR of ZEN as 100%. - No data. CR: cross reactivity.

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
