# Peer review of "Preparation and Characterization of Monoclonal Antibodies with High Affinity and Broad Class Specificity against Zearalenone and Its Major Metabolites"

_toxins, 2021, doi:10.3390/toxins13060383_

Round 1
Reviewer 1 Report
This manuscript describes the development of monoclonal antibodies against zearalenone. It described an interesting study but lacks some details and clarifications. After major revision, I would be happy to recommend it published in Toxins.
- The paper describes production of monoclonal antibodies in vivo by the ascites method. Yet, the methodology remains debatably acceptable. For example, the The European Union Reference Laboratory for alternatives to animal testing (EURL ECVAM) has clearly stated that “Antibody production using the ascites method should no longer be acceptable under any circumstances” (Barroso et al. EURL ECVAM Recommendation on Non-Animal-Derived Antibodies). Also, the American Anti-Vivisection Society (AAVS) of USA has appealed the NIH to prohibit the use of animals in the production of mAbs (National Research Council, 1999). Could the authors discuss this ethical concern? Is it necessary to use the ascites method in their work, or could the antibodies be produced in vitro?
- Please refer to Figure 12 when discussing the OEA method in the main text. Now, this figure appears only in the experimental section and thus the reader might miss it.
- Line 129–130: What do you mean with the BSA method.
- Line 132: Please rephrase the sentence “An indication that immunogen ZEN-BSA contains BSA”. It is not complete and not clear, what the authors want to say.
- Line 147–148: “The hapten-carrier effect theory states that the immune response depends on the physicochemical properties of the immunogen and the immunization method.” Please provide a reference.
- Some abbreviations are explained only in the abstract. They should be included also in the main text where they are mentioned for the first time (e.g., IR, inELISA and icELISA, CR)
- Lines 228–229: I don’t agree with the authors about the definition of immunoassay sensitivity. The IC50 value can be used as measured to evaluate the immunoassay performance but, in fact, several things contribute to the immunoassay sensitivity, including but not limited to, antibody affinity to the analyte (and affinity of the competitor in competitive immunoassays), the reporter and their detectability, the experimental and measurement error, and non-specific binding.
- In addition to (instead of) Figure 9, it would be interesting for the reader to see the actual calibration curves in the presence of methanol, not just the Amax values. Similarly, the effect of the pH in Figure 10, might be better seen if the authors showed also the original calibration curves and would thus provide more complete data for the reader.
- Please include the error bars in the figures were replicates were used.
- Figure 11: What is the black line? If it does not correspond to any fitted curve, please consider removing it.
- Please describe also the CR study and the results with other mycotoxins in the main text.
- Please check the number formatting in the text, I believe some numbers in the superscript have been incorrectly formatted, e.g., line 22 about the affinity constants 4.15×109 and 6.54×109 L/mol, and line 131 1700 cm-1. Similarly, the subscripts have disappeared form the chemicals, e.g. Na2HPO4·12H2O
- Lines 39–40: Please write the species and genus names in italics.
- Figure 3: The explanation for the green line is missing in the figure legend.
- Figure 4: y-axis title has a typo, should be “Absorbance” (also in Figures 6 and 8) and the x-axis is missing the concentration unit (if the dilution is plotted, label the axis “dilution of pAb” instead of “diluted concentration pAb”).
- Use the same units everywhere when discussing the IC50 for example (either µg/L or ng/mL).
- Consider revising the number of significant digits in Table 1 (unless the precision of your methods is actually this good).
- Line 248: “the highest IC50 value”, do you mean the lowest?
- Line 285: “Methanol is widely used in ELISA as an organic cosolvent to detect mycotoxins”, methanol is rather used to extract the mycotoxins from the food matrix than actually for the detection.
Author Response
This manuscript describes the development of monoclonal antibodies against
zearalenone. It described an interesting study but lacks some details and clarifications. After major revision, I would be happy to recommend it published in Toxins.
Thank you for your affirmation and encouragement of the manuscript. According to your suggestions, we have made major revisions and tried to pay attention to details and clarification in the revisions.
The paper describes production of monoclonal antibodies in vivo by the ascites
method. Yet, the methodology remains debatably acceptable. For example, the The European Union Reference Laboratory for alternatives to animal testing (EURLECVAM) has clearly stated that “Antibody production using the ascites method should no longer be acceptable under any circumstances” (Barroso et al. EURLECVAM Recommendation on Non-Animal-Derived Antibodies). Also, the American Anti-Vivisection Society (AAVS) of USA has appealed the NIH to prohibit the use of animals in the production of mAbs (National Research Council, 1999). Could the authors discuss this ethical concern? Is it necessary to use the ascites method in their work, or could the antibodies be produced in vitro?
Thanks for your valuable advice. According to your suggestion, we have discussed this issue and marked them in red on lines 200-212, hoping that you can accept our point of view.
Please refer to Figure 12 when discussing the OEA method in the main text. Now, this figure appears only in the experimental section and thus the reader might miss it.
Thanks for your valuable advice. According to your suggestion, We have made a
change in the text discussing the OEA method, marked in red on line 283.
Line 129–130: What do you mean with the BSA method.
Thanks for your valuable advice, we are sorry for our carelessness. According to your suggestion, we have checked the manuscript carefully and corrected in red on line 155.
Line 132: Please rephrase the sentence “An indication that immunogen ZEN-BSA
contains BSA”. It is not complete and not clear, what the authors want to say.
Thanks for your valuable advice, we are sorry for our explanation confused you.
According to your suggestion, we have replaced the sentence on the line 157 of
revised marked manuscript with red.
Line 147–148: “The hapten-carrier effect theory states that the immune response
depends on the physicochemical properties of the immunogen and the immunization method.” Please provide a reference.
Thanks for your valuable advice, we are sorry for our explanation confused you. Our understanding of the hapten-carrier effect theory is wrong, therefore, the statement of the sentence is incorrect. We have revised it and added references, and marked it in red on lines 117-118. We hope you can accept our views.
Some abbreviations are explained only in the abstract. They should be included also in the main text where they are mentioned for the first time (e.g., IR, inELISA and icELISA, CR)
Thanks for your valuable advice, we are sorry for our carelessness. According to your suggestion, we have carefully checked and revised the manuscript.
Lines 228–229: I don’t agree with the authors about the definition of immunoassay sensitivity. The IC50 value can be used as measured to evaluate the immunoassay performance but, in fact, several things contribute to the immunoassay sensitivity, including but not limited to, antibody affinity to the analyte (and affinity of the competitor in competitive immunoassays), the reporter and their detectability, the experimental and measurement error, and non-specific binding.
Thank you for your valuable guidance. Our understanding of IC50 value and
sensitivity are not profound and comprehensive enough, which led to the wrong
definition of sensitivity for immunoassay. We have revised it and marked them in red on lines 245-247.
In addition to (instead of) Figure 9, it would be interesting for the reader to see the actual calibration curves in the presence of methanol, not just the Amax values. Similarly, the effect of the pH in Figure 10, might be better seen if the authors showed also the original calibration curves and would thus provide more complete data for the reader.
Thanks for your valuable advice. According to your suggestion, we have replaced the previous Figure 9 and Figure 10 with the new Figure 9 and Figure 10 in the line 311-314 of revised marked manuscript with red.
Please include the error bars in the figures were replicates were used.
Thanks for your valuable advice.
According to your suggestion, we have introduced error bars in all figures with repeated experimental data, including line 184 Figure 4, line 187 Figure 5, line 228 Figure 6, line 231 Figure 7, line 242 Figure 8, line 311 Figure 9 and line 314 Figure 10.
Figure 11: What is the black line? If it does not correspond to any fitted curve, please consider removing it.
Thanks for your valuable advice. According to your suggestion, we have deleted the red line in Figure 11 on line 318, which is the trend line of the fitted curve.
Please describe also the CR study and the results with other mycotoxins in the main text.
Thanks for your valuable advice. According to your suggestion, we have described the results of the CR study and the results with other mycotoxins in red on lines 267-270 of the text.
Please check the number formatting in the text, I believe some numbers in the
superscript have been incorrectly formatted, e.g., line 22 about the affinity constants 4.15×109 and 6.54×109 L/mol, and line 131 1700 cm-1. Similarly, the subscripts have disappeared form the chemicals, e.g. Na2HPO4·12H2O
Thanks for your valuable advice. We are sorry for our error in the number formatting in the text. According to your suggestion, we have carefully reviewed and revised, marked in red, including lines 19, 21, 157, 159, 362, 364, 369, 379, 480 and 496.
Lines 39–40: Please write the species and genus names in italics.
Thanks for your valuable advice. We are sorry for our error description. According to your suggestion, we have written out the names of species and genus in italics in the lines 39-41 of revised marked manuscript with red.
Figure 3: The explanation for the green line is missing in the figure legend.
Thanks for your valuable advice.
According to your suggestion, We have already added it in red on line 172.
Figure 4: y-axis title has a typo, should be “Absorbance” (also in Figures 6 and 8) and the x-axis is missing the concentration unit (if the dilution is plotted, label the axis “dilution of pAb” instead of “diluted concentration pAb”).
Thanks for your valuable advice. We are sorry for our carelessness and spelling
mistakes. According to your suggestion, we have carefully reviewed and revised,
marked in red, including line 172 Figure 3, line 184 Figure 4, line 228 Figure 6, line 242 Figure 8.
Use the same units everywhere when discussing the IC50 for example (either μg/L or ng/mL).
Thanks for your valuable advice. According to your suggestion, we have revised it and marked in red on line 180.
Consider revising the number of significant digits in Table 1 (unless the precision of your methods is actually this good).
Thanks for your valuable advice. According to your suggestion, we have revised and marked them in red on line 189 Table 1.
Line 248: “the highest IC50 value”, do you mean the lowest?
Thanks for your valuable advice. We are sorry for our carelessness, and have revised “the highest” to “the lowest” in red on line 266.
Line 285: “Methanol is widely used in ELISA as an organic cosolvent to detect
mycotoxins”, methanol is rather used to extract the mycotoxins from the food matrix than actually for the detection.
Thanks for your valuable guidance. Our understanding and description of the
application and function of methanol in ELISA kit are not accurate enough.
According to your suggestion, we have revised this sentence in red on line 296.
Reviewer 2 Report
Authors have done huge among of works, but the messages are not clear. Many information should be introduction for example methods for section of snythesis, immunization method. There are gaps to introduce (1) FTIR, (2) inELISA and icELISA, (3) the coating antigen.
line 97 to line 128 are related with literature. The content should be moved to introduction.
line 129 it should be introduction of why you are doing FTIR.
Line 147-158 are related with literature. The content should be moved to introduction.
Line 179-185 Line 147-158 are related with literature. The content should be moved to introduction.
It is difficult to understand so many information without correct format and order.
some minor errors
Line 19 The mAbs titers were between (2.56 to 5.12) ×102 in supernatants and (1.28 to 5.12) ×105 .
please indicate which peak are hydroxyl and esters in figure 2.
Please cite the green curve in the figure 3
line 131 should be 1700 cm-1
line 133 should be 2100–2300 cm-1
Author Response
line 97 to line 128 are related with literature. The content should be moved to
introduction.
Thanks for your valuable advice. According to your suggestion, we have slightly
revised this section and moved it to the introduction, marked in red on lines 89-116.
line 129 it should be introduction of why you are doing FTIR.
Thanks for your valuable advice. According to your suggestion, we explained why IR is doing to identify the immunogen ZEN-BSA, and the revised part is marked in red on lines 149-153.We hope you will accept our explanation.
Line 147-158 are related with literature. The content should be moved to introduction.
Thanks for your valuable advice. According to your suggestion, we have slightly
revised this section and moved it to the introduction, marked in red on lines 117-131.
Line 179-185 Line 147-158 are related with literature. The content should be moved to introduction.
Thanks for your valuable advice. According to your suggestion, we have slightly
revised this section and moved it to the introduction, marked in red on lines 132-139.
It is difficult to understand so many information without correct format and order.
Thanks for your valuable guidance, we are sorry for the incorrect format and order of our manuscript. According to your suggestion, we have revised the format and order of the manuscript.
some minor errors
Line 19 The mAbs titers were between (2.56 to 5.12) ×102 in supernatants and (1.28 to 5.12) ×105 .7
Thanks for your valuable advice, we are sorry for our carelessness. According to your suggestion, we have revised the formats to superscript in the line 19 and marked with red.
please indicate which peak are hydroxyl and esters in figure 2.
Thanks for your valuable advice, we are sorry for our carelessness. According to your suggestion, we have revised this sentence and indicated which peak are hydroxyl and esters in Figure 2, marked in red on line 159.
Please cite the green curve in the figure 3
Thanks for your valuable advice. According to your suggestion, We have already
added it in red on line 166.
line 131 should be 1700 cm-1
Thanks for your valuable advice, we are sorry for our carelessness. we have revised the formats to superscript in the line 156 and marked with red.
line 133 should be 2100–2300 cm-1
Thanks for your valuable advice, we are sorry for our carelessness. we have revised the formats to superscript in the line 159 and marked with red.
Round 2
Reviewer 1 Report
Thank you for the careful revision of the manuscript and the answers to my comments. The manuscript has improved but it still lacks attention to the detail and, in my opinion, requires minor corrections before publishing.
I acknowledge the authors for including some discussion regarding the ethics of the ascites methods. However, it still remains unclear to the reader why the ascites method was used in this study if the authors truly believe that the method should not be used. Could you comment on this?
Manuscript still contains several minor spelling and formatting mistakes, e.g. missing spaces and species names without the italics, as well as strange sentences, such as on line 109 “However, none of these methods satisfied was in tandem with our expected outcome.” Please revise the entire manuscript carefully to correct the grammar mistakes.
Please include the explanation of the green curve in the legend of Figure 3 where you explain the black and red curves as well.
Include the explanation of the green and violet curves in the legend of Figure 6.
Correct Figure 7 caption “The titers The IC50 values”, which one do you mean?
Figure 8, what do you mean with “Affinity constant (Kaff) curves”?
Revise figures 9 and 10. The formatting is very bad and insets are placed randomly on top of the actual graph. Also at the end of page 13, there is some weird figure of the Amax/IC50 (inset of figure 9?) which I think does not belong here.
